# Suppression of interferon signaling via small-molecule modulation of TFAM

Dionisia Sideris[1], Husan Lee[1], Lyndsay Olson[1], Kalyan Nallaparaju[1], Keiichiro Okuyama[2], Jeffrey Ciavarri[2], Robert Lafyatis[3], Mads Larsen[4], Bo Lin[4], Irene Alfaras[4], Jason Kennerdell[4], Toren Finkel[4,5,6], Yuan Liu[4,6,7], Bill Chen[4,6,7,8], Lin Lyu[1]*

[1]Astellas Pharmaceuticals Inc, Oncology Research, Cancer Biology, Cambridge, United States; [2]Astellas Pharmaceuticals Inc, Oncology Medicinal Chemistry, Cambridge, United States; [3]Division of Rheumatology and Clinical Immunology, University of Pittsburgh, Pittsburgh, United States; [4]Aging Institute, University of Pittsburgh/UPMC, Pittsburgh, United States; [5]Department of Medicine, Division of Cardiology, University of Pittsburgh, Pittsburgh, United States; [6]Department of Medicine, Division of Pulmonary, Allergy and Critical Care Medicine, University of Pittsburgh, Pittsburgh, United States; [7]Generian Pharmaceuticals, Pittsburgh, United States; [8]Vascular Medicine Institute, University of Pittsburgh, Pittsburgh, United States

*For correspondence:
lin.lyu@astellas.com

## eLife Assessment

Using high-throughput small-molecule screening, this study discloses novel modulators of the mitochondrial transcription factor A (TFAM), a key regulator of mitochondrial function. Reviewers viewed the targeting of TFAM as innovative and the study's conclusions as potentially **important** (especially the effects on inflammation). However, the lack of evidence for a direct effect of the compounds on TFAM activity weakens the paper's key conclusion and renders the study **incomplete**.

**Abstract** The mitochondrial transcription factor A (TFAM) is essential for mitochondrial genome maintenance. It binds to mitochondrial DNA (mtDNA) and determines the abundance, packaging, and stability of the mitochondrial genome. Because its function is tightly associated with mtDNA, TFAM has a protective role in mitochondrial diseases, and supportive studies demonstrate reversal of disease phenotypes by TFAM overexpression. In addition, TFAM deficiency has been shown to cause release of mtDNA into the cytosol and activation of the cGAS/STING innate immune response pathway. As such, TFAM presents as a unique target for therapeutic intervention, but limited efforts for activators have been reported. Herein, we disclose novel TFAM small-molecule modulators with sub-micromolar activity. Our results demonstrate that these compounds result in an increase of TFAM protein levels and mtDNA copy number. This results in inhibition of a mtDNA stress-mediated inflammatory response by preventing mtDNA escape into the cytosol. Furthermore, we see beneficial effects in cellular disease models in which boosting TFAM activity has been advanced as a disease-modifying strategy including improved energetics in MELAS cybrid cells and a decrease of fibrotic markers in systemic sclerosis fibroblasts. These results highlight the therapeutic potential of using small-molecule TFAM activators in indications characterized by mitochondrial dysfunction.

## Introduction

Mitochondrial transcription factor A (TFAM) is a nuclear-encoded transcription factor essential for determining the abundance and stability of mitochondrial DNA (mtDNA). It plays a central role in packaging mtDNA into nucleoprotein structures called nucleoids, with each nucleoid containing a single copy of mtDNA compacted by TFAM. These nucleoids can be categorized based on their activity: active nucleoids drive mtDNA transcription and replication, while inactive nucleoids likely serve as a storage pool to protect mtDNA from damage or release. The compaction level of mtDNA nucleoids, determined by the TFAM:mtDNA ratio, plays a critical role in regulating mitochondrial gene expression. Excessive TFAM relative to mtDNA leads to hyper-compaction of nucleoids, which represses transcription and replication (*Kaufman et al., 2007*; *Miranda et al., 2022*; *Sánchez-Quintero et al., 2023*). As a result, TFAM plays a crucial role in mitochondrial biogenesis by facilitating mtDNA transcription and replication to sustain mitochondrial gene expression and protect mtDNA from damage. Through these functions, TFAM ensures mitochondrial genome stability and supports the generation and maintenance of functional mitochondria (*Miranda et al., 2022*).

The TFAM:mtDNA ratio can be influenced by several transcriptional as well as post-translational mechanisms, including changes in TFAM expression, increased TFAM degradation via the Lon protease, and mtDNA copy number alterations. For example, in response to cellular environmental cues, TFAM is upregulated by PGC1-a to stimulate an increase in mtDNA copy number and mitochondrial biogenesis (*Wu et al., 1999*). Additionally, serine phosphorylation of TFAM by cAMP-dependent protein kinase (PKA) impairs its DNA-binding ability, rendering it more susceptible to Lon-mediated degradation (*Lu et al., 2013*). These regulatory mechanisms highlight the importance of precise TFAM level control in preserving mitochondrial function. In vivo studies further supported the idea that modulating TFAM expression could be a promising therapeutic strategy to counteract mitochondrial dysfunction. Studies using cellular and animal models have shown that modulation of TFAM levels to stimulate an increase in mtDNA copy number by 1.5-fold provides a safe and beneficial intervention, whereas strong overexpression of TFAM may have a detrimental effect and result in excessive repression of mitochondrial gene expression, nucleoid clustering, and ultrastructural changes of mitochondria in different tissues (*Bonekamp et al., 2021*).

Beyond its fundamental role in mitochondrial maintenance, TFAM has significant therapeutic potential for disorders involving mtDNA instability and mitochondrial dysfunction. One promising application is preventing mtDNA escape into the cytoplasm, a pathological phenomenon linked to inflammation and organ damage. For instance, in kidney fibrosis—a common pathway leading to end-stage renal failure—it has been observed that significant mitochondrial defects including the loss of TFAM in kidney tubule cells results in escape of mtDNA into the cytosol. This triggers activation of the cytosolic cGAS-STING pathway and an elevated type I interferon response (*West et al., 2015*).

In addition to fibrosis, mtDNA escape activates multiple inflammatory pathways, including cGAS-STING signaling, NLRP3 inflammasome activation, and TLR9-mediated NF-κB signaling. These pathways collectively contribute to a vicious cycle of mitochondrial damage and immune activation, which underlies conditions such as neuroinflammation and autoimmune disorders (*Lin et al., 2022*). Preclinical and clinical studies further reported that elevated levels of circulating mtDNA have been observed in patients with ulcerative colitis and Crohn's disease, and immune activation has been identified as a causal factor in the pathogenesis of mitochondrial diseases, which indicated that targeting immune pathways in mitochondrial diseases can provide significant therapeutic benefits (*Sánchez-Quintero et al., 2023*).

Stabilizing TFAM to prevent mtDNA escape, increase mtDNA copy number, and enhance mitochondrial function may be a compelling therapeutic strategy. Based on this rationale, we hypothesize that small-molecule modulators of TFAM could serve as first-in-class disease-modifying treatments for conditions characterized by leaky mtDNA. Herein, we employed a cellular thermal shift assay (CESTA) to identify novel TFAM small-molecule modulators. Follow-up analog synthesis resulted in compounds that stabilized TFAM at sub-micromolar levels. Our results show that these compounds lead to an increase in TFAM protein levels, which prevents mtDNA escape into the cytosol, thereby inhibiting a mtDNA stress-mediated inflammatory response. Furthermore, using the compounds in cellular disease models, we observed increased ATP levels in MELAS cybrid cells and a decrease in fibrotic markers in Systemic Sclerosis fibroblasts. This study highlights the therapeutic potential of

using small-molecule TFAM activators in diseases characterized by mitochondrial dysfunction, paving the way for future translational and clinical applications.

## Results

### Discovery of small-molecule modulators of TFAM

We adapted a previously described approach to use an in-cell thermal shift assay (CETSA) to identify small-molecule binders to a protein of interest (*Martinez et al., 2018*). In this report, we describe our discovery and characterization of TFAM activators identified from a ChemDiv screening library (*Figure 1A*).

A total of 160 hits were selected from the screen that changed the thermal shift of TFAM (*Figure 1B*) and were further evaluated in our screening funnel which included measuring TFAM protein levels using a HiBiT CRISPR cell line (data not shown). Because TFAM levels are known to directly control mtDNA copy number (*Larsson et al., 1998*; *Matsushima et al., 2003*), we further profiled the hits that effected an increase in TFAM levels in a mtDNA assay (*Figure 1C*).

From these efforts, we identified three unique chemotypes that increased TFAM protein levels and altered mtDNA copy number, while exhibiting minimal cytotoxic effects (*Figure 1C*). Within this set of compounds, compound **1** appeared to inhibit TFAM activity as demonstrated by a decrease in mtDNA copy number. Conversely, treatment of cells with compound **2** or compound **15** increased mtDNA copy number. Compound **2**, an arylsulfonamide, was selected as the lead series candidate for further validation by confirming the effect on mtDNA copy number in multiple cell lines including A549, 22RV1, A172, and T47D cells. A consistent increase in the mtDNA levels was observed upon treatment of the cells with compound **2**, with T47D cells showing an average twofold increase at 6.6 uM. Overall, our results suggest that our screening strategy can identify compounds that positively or negatively influence TFAM and mtDNA.

### TFAM modulators suppress interferon signaling pathway

To further validate that our compounds modulate TFAM activity, we interrogated their impact on the downstream signaling cascade mediated by the cGAS-STING pathway. TFAM is known to stabilize mtDNA, and its deficiency leads to the release of mtDNA into the cytosol (*West et al., 2015*). This release activates the cGAS-STING pathway, which subsequently triggers the downstream expression of interferon-stimulating genes (ISG) and promotes type-I IFN production. Therefore, compounds that stabilize and activate TFAM should prevent mtDNA release and limit IFN production.

To test this hypothesis, we established and validated an in-house assay to interrogate our compounds' ability to suppress downstream cytokine production, using AlphaLisa to measure CXCL10 as a representative marker. To mimic a cellular environment with mitochondrial damage, we used TNF-α, previously shown to induce a cGAS-STING-dependent interferon response by altering mitochondrial function and increasing cytosolic mtDNA levels (*Willemsen et al., 2021*). Toward this end, THP-1 cells were treated for 24 h and 48 h with TNF-α resulting in a time- and dose-dependent increase in secreted CXCL10 (*Figure 2—figure supplement 1A*). Subsequent treatment with the known covalent STING inhibitor H151 (*Hu et al., 2022*) abolished CXCL-10 production (*Figure 2—figure supplement 1B*), confirming that TNF-α induced CXCL10 is dependent on the cGAS/STING pathway. Furthermore, treatment with the VDAC inhibitor VBIT-4, which is known to block mtDNA release (*Kim et al., 2023*) by inhibiting the oligomerization and opening of the voltage-dependent anion channel (VDAC) pore, also suppressed the production of CXCL10 in our assay (*Figure 2—figure supplement 1C*), providing additional evidence that cGAS-STING activation is triggered by mtDNA release driven by TNF-α treatment. These results indicate that this assay can be used as an indirect method to interrogate the impact of TFAM modulators in suppressing mtDNA release and activation of cGAS-STING pathway.

We then assessed the effect of our novel TFAM modulating compounds on CXCL-10 secretion using the method shown in *Figure 2A*. As expected, THP1 cells treated with TNF-α for 48 h significantly stimulated production of CXCL-10; however, pretreatment with compounds **2**, **3** (a close analog of compound **2** acquired from ChemDiv as part of a hit expansion effort; *Figure 3A*), or **15** for 48 h inhibited CXCL10 secretion in a dose-dependent fashion (*Figure 2A and B*). These results demonstrate that compound pretreatment significantly reduced TNF-α induced cytokine production. Given

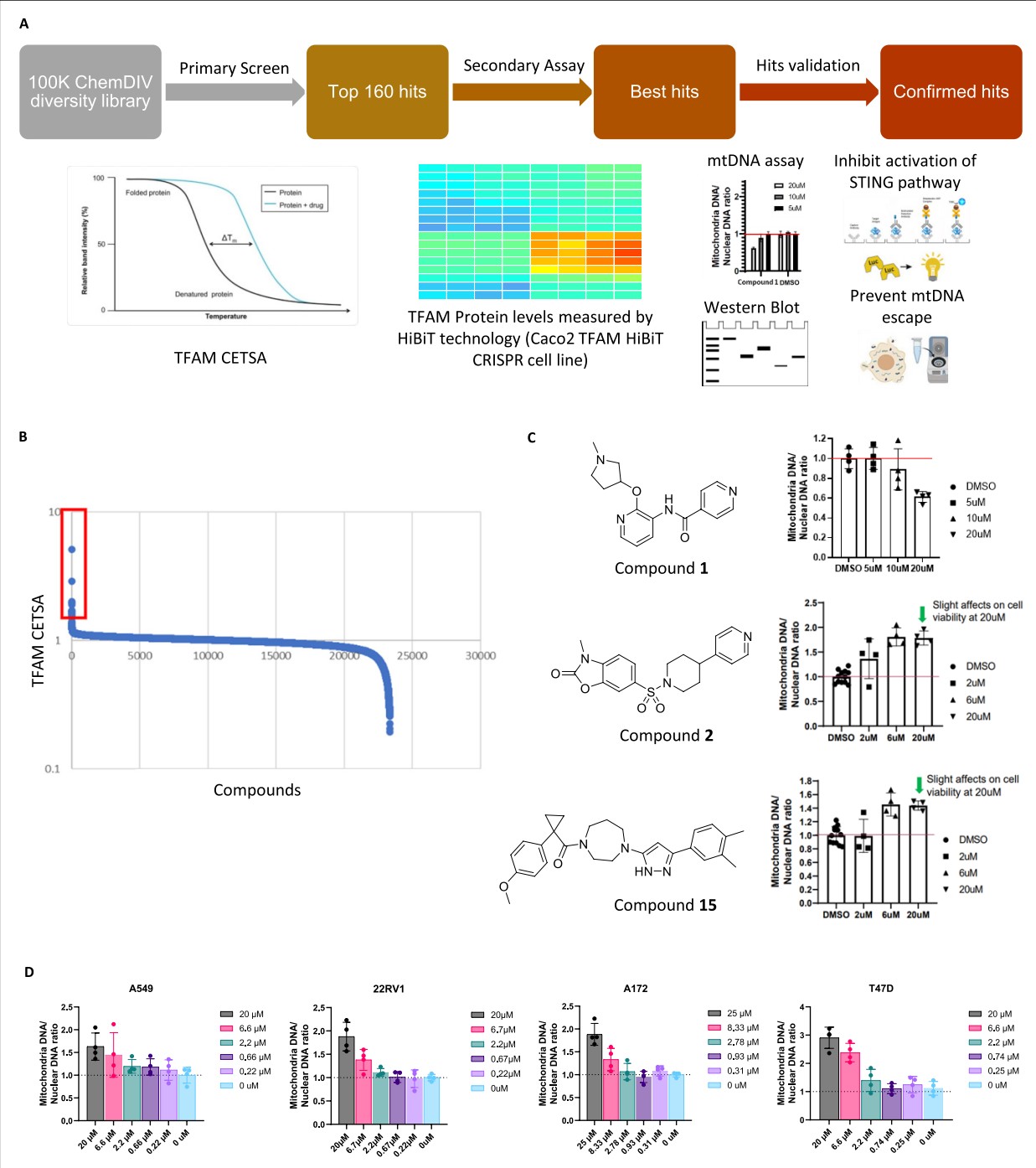

**Figure 1.** Development of a high-throughput screen for the identification of potent small-molecule activators to TFAM. (**A**) Diagram of primary screen and secondary assays (CETSA, stabilizing TFAM to increase mtDNA copy number, prevent escape and inhibit activation of STING pathway). (**B**) Results from CETSA screen. (**C**) Hit validation in mtDNA copy number assay—compounds **1**, **2**, and **15** evaluated at increasing concentrations in HeLa cells. (**D**) Compound **2** evaluated at increasing concentrations across multiple cell lines. Figures are representatives of at least 2 independent experiments. Graph shows one representative experiment of two independent experiments. Error bars represent ± SD from n=4 biological replicates. Source data for this figure are available in *Figure 1—source data 1* (raw data and analysis).

The online version of this article includes the following source data for figure 1:

**Source data 1.** Raw data and analysis results to generate the graphs shown in *Figure 1*.

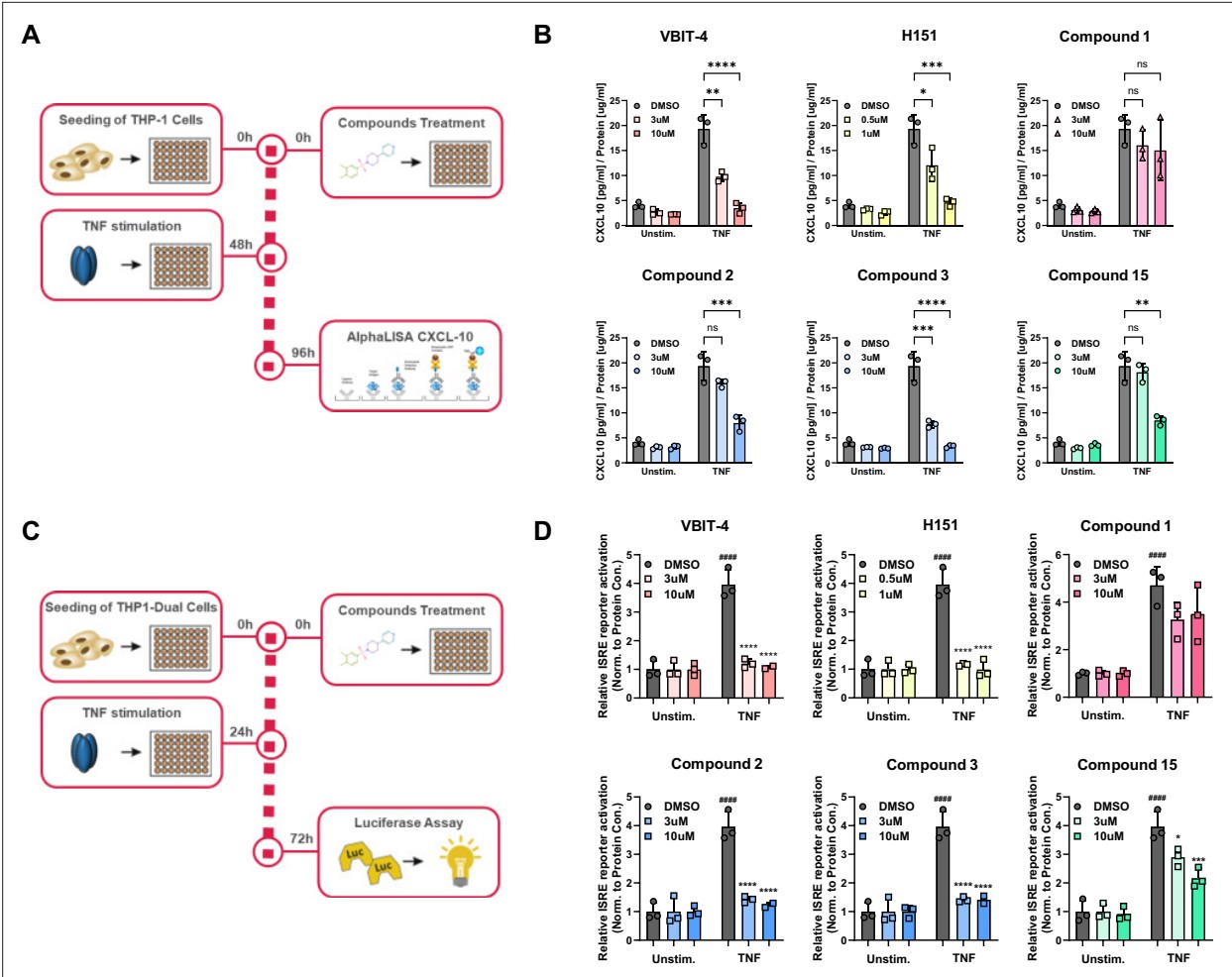

**Figure 2.** Compounds 2, 3, and 15 repress ISG signaling mediated by mtDNA damage and the cGAS/STING pathway. (**A**) Flowchart of the CXCL-10 experiment. (**B**) Compounds 2, 3, and 15 repressed ISG signaling in CXCL-10 production. THP-1 cells were pre-treated with individual compounds 48 h prior to 100 ng/mL TNFa stimulation for 48 h. CXCL-10 levels were measured. (**C**) Flowchart of the ISRE experiment. (**D**) Compounds 2, 3, and 15 repressed ISG signaling in IRF3 production. THP-1-Dual cells were pre-treated with individual compounds 24 h prior to 100 ng/mL TNFa stimulation for 48 h. The inhibition of type I IFN response was monitored via ISRE reporter activation. Figures are representatives of at least two independent experiments. Graph shows one representative experiment of two independent experiments. Error bars represent ± SD from n=3 biological replicates. *p<0.05; **p<0.01; ***p<0.001; ****p<0.0001. Source data for this figure are available in **Figure 2—source data 1** (raw data and analysis).

The online version of this article includes the following source data and figure supplement(s) for figure 2:

**Source data 1.** Raw data and analysis results to generate the graphs shown in **Figure 2** .

**Figure supplement 1.** TNFa stimulates upregulation of CXCL10 protein levels.

**Figure supplement 2.** Assessment of compound assay interference.

that these compounds were shown to increase TFAM and mtDNA levels, these findings suggest that their effect on cytokine suppression may be linked to enhanced mtDNA stability.

To further validate the effect of our compounds in the cGAS-STING signaling pathway, we used an orthogonal ISRE reporter activation assay, where interferon-stimulated gene (ISG) expression is assessed by secreted luciferase activity (**Figure 2C**). Treatment of cells with TNF-α for 48 h significantly increased ISRE activity, while pre-treatment with VBIT-4 (10 uM) abrogated the signal (**Figure 2D**). Similarly, pre-treatment with compounds **2**, **3**, or **15** reduced ISRE activity in a dose-dependent manner (**Figure 2D**), while compound **1**, previously identified as a likely inhibitor of TFAM, had no effect on ISRE activity.

To ensure that observed effects were not due to a non-specific inhibition of luciferase substrate activity, a control experiment was performed in which compounds were added to cells 5 min prior to

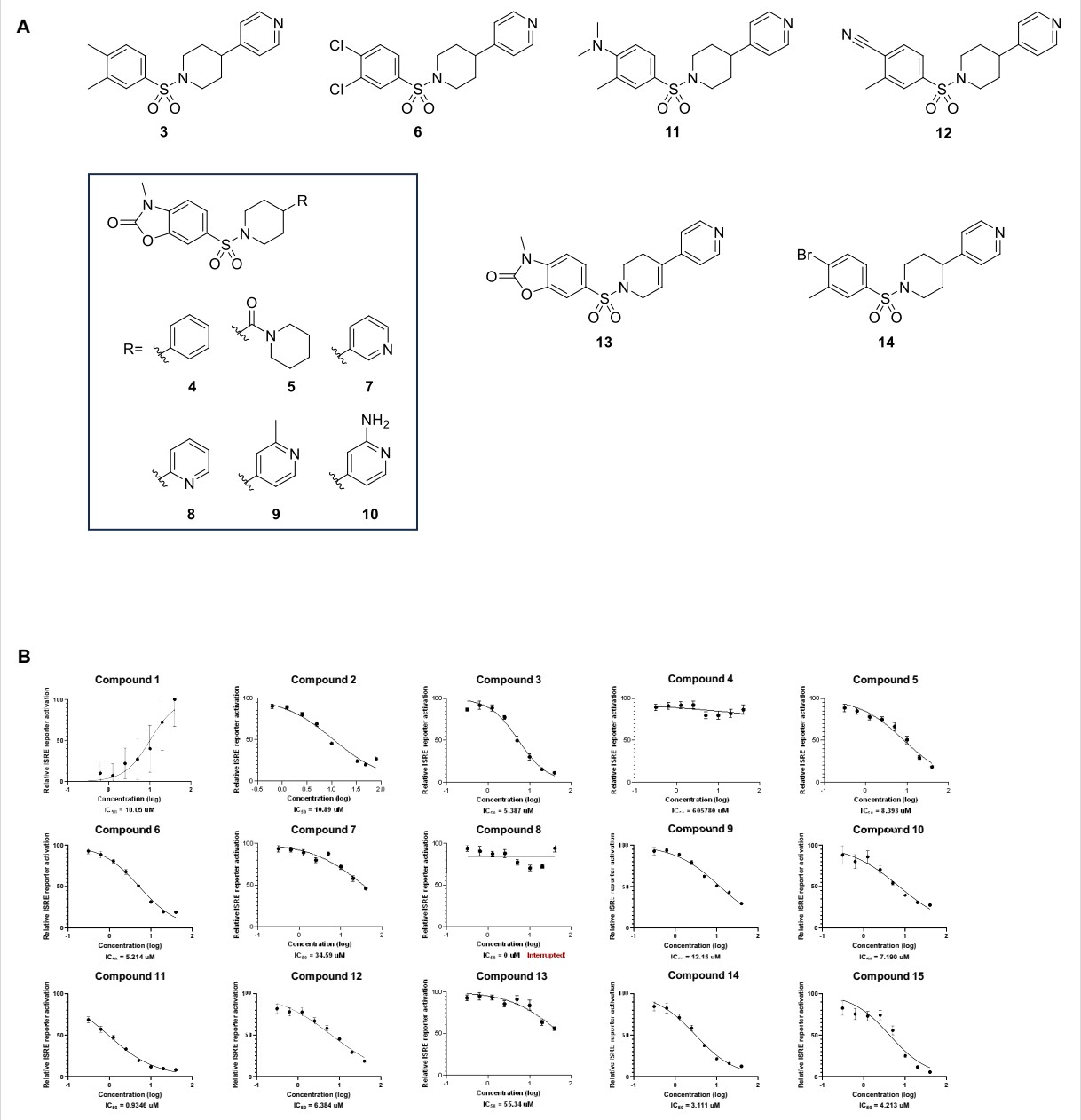

**Figure 3.** Hit expansion – compound **11** identified as a TFAM activator with sub-micromolar potency. (**A**) Chemical structures of hit compounds and related analogs. (**B**) TFAM modulators were profiled in the ISRE assay and display dose-dependent suppression of ISG signaling. THP-1-Dual cells were pre-treated with individual compounds 24 h prior to 100 ng/mL TNFa stimulation for 48 h. ISRE reporter activation was measured. Figures are representatives of at least two independent experiments. Graph shows one representative experiment of two independent experiments. Error bars represent ± SD from n=3 biological replicates. *p<0.05; **p<0.01; ***p<0.001; ****p<0.0001. Source data for this figure are available in *Figure 3—source data 1* (raw data and analysis).

The online version of this article includes the following source data for figure 3:

**Source data 1.** Raw data and analysis results to generate the graphs shown in *Figure 3*.

assay detection. Using this approach, no effect on ISRE luciferase signal was detected, confirming that the observed decrease in cytokine production and interferon signaling was mediated via compound modulation of TFAM and not by interfering with the luciferase signal (*Figure 2—figure supplement 2*).

To further explore structure-activity relationships, analogs of the arylsulfonamide hit (compound **2**) were synthesized and evaluated in the ISRE assay (*Figure 3A*). Investigation of 2- and 3-pyridyl substitution (compounds **7** and **8**) in place of the 4-pyridyl moiety of **2** resulted in abrogation of activity as did replacement with a phenyl ring (compound **4**). Comparable activity was observed with either a methyl or amine substituent adjacent to the pyridyl nitrogen (compounds **9** and **10**). Interestingly, activity was maintained with the incorporation of a piperidyl amide group (compound **5**). Holding the 4-pyridyl constant and interrogating functionality alternative to the oxazolidinone moiety, electron-withdrawing and electron-donating groups were incorporated into the molecule (compounds **3**, **6**, **11**, **12**, and **14**). Of these analogs, compound **11** displayed the greatest inhibitory activity (IC$_{50}$ 0.9 uM). These results suggest emerging SAR with the potential to further improve potency in a lead optimization effort.

## TFAM modulators increase TFAM protein levels and prevent mtDNA cytosolic escape

To further prove that ISG suppression is due to TFAM modulation, we examined cGAS, a cytosolic sensor that detects mitochondrial and cytosolic DNA, activating the STING pathway through 2'–3' cGAMP synthesis in the cytosol. We wanted to further validate that our compounds repress ISG signaling that specifically resulted from mitochondrial damage. To test this, we employed cGAMP instead of TNF-α to activate ISG signaling. H151, a cytosolic STING inhibitor, blocked the ISRE reporter activation triggered by cGAMP while ISRE activity was not affected by VDAC inhibitor VBIT-4 as expected (*Figure 4A*). Moreover, none of our compounds repressed ISG signaling that was a direct result of cytosolic STING activation (*Figure 4A*), suggesting that the compounds are exerting their function in the mitochondria rather than directly on cGAS-STING.

In addition, we tested the effect of our novel arylsulfonamide analog series on TFAM protein levels in T47D cells (*Figure 4B*). Compared with untreated cells, TFAM protein levels significantly increased in the presence of compounds **2**, **3**, or **11** in a dose-dependent manner, reaching approximately three-fold, fivefold, and twofold increases, respectively, at the highest concentration tested (*Figure 4B*). H151 and compound **1** had no effect on protein levels as expected. Unexpectedly, compound **15** also failed to increase TFAM protein levels, suggesting that this compound may activate TFAM through other mechanisms (*Figure 4B*). Importantly, in contrast to the increase in protein levels, TFAM mRNA levels did not show significant changes upon exposure to these analogs, suggesting that TFAM increase in protein levels was not a result of transcription activation (*Figure 4C*).

We sought to gain confidence that our observed repression of ISG signaling is a result of increased TFAM protein levels. To test this, we reduced cellular TFAM protein levels by siRNA transfection (*Figure 4D*). Notably, downregulation of TFAM significantly attenuated the ability of VBIT-4, compounds **2** or **3**, to repress ISG signaling (*Figure 4D*) but did not impact the activity of either H151 or compound **15**. This finding is consistent with WB results that compound **15** does not upregulate the levels of TFAM and is likely acting through other mechanisms.

We further investigated whether our TFAM compounds could inhibit mtDNA stress in THP-1 cells, which is characterized as a release of mtDNA into the cytosol. We used cellular fractionation to assess the relative amount of mtDNA escaping in the cytosol after TNF treatment. As shown in *Figure 4E*, TNF-α treatment led to an increase in the levels of cytosolic mtDNA, but treatment with compound **3** significantly attenuated cytosolic mtDNA release in a dose-dependent manner. In contrast, cytosolic mtDNA levels were not significantly affected by compound **15** (*Figure 4E*). These results provide evidence that our arylsulfonamide compounds are acting on the ISG pathway by directly modulating the levels of TFAM and the stability of mtDNA, while compound **15** might have been affecting ISG signaling through other mechanisms.

## TFAM modulators can increase ATP, reduce fibrotic markers, and enhance Treg function in cellular disease models

As our work demonstrates, an increase in TFAM protein levels by way of chemical intervention can lead to stabilization of mtDNA and prevent its release into the cytosol. This can have an important therapeutic effect on mitochondrial-related ailments.

For example, mitochondrial encephalo-myopathy, lactic acidosis, and stroke-like episodes (MELAS) syndrome typically results from point mutations in tRNA genes encoded by mtDNA, resulting in an

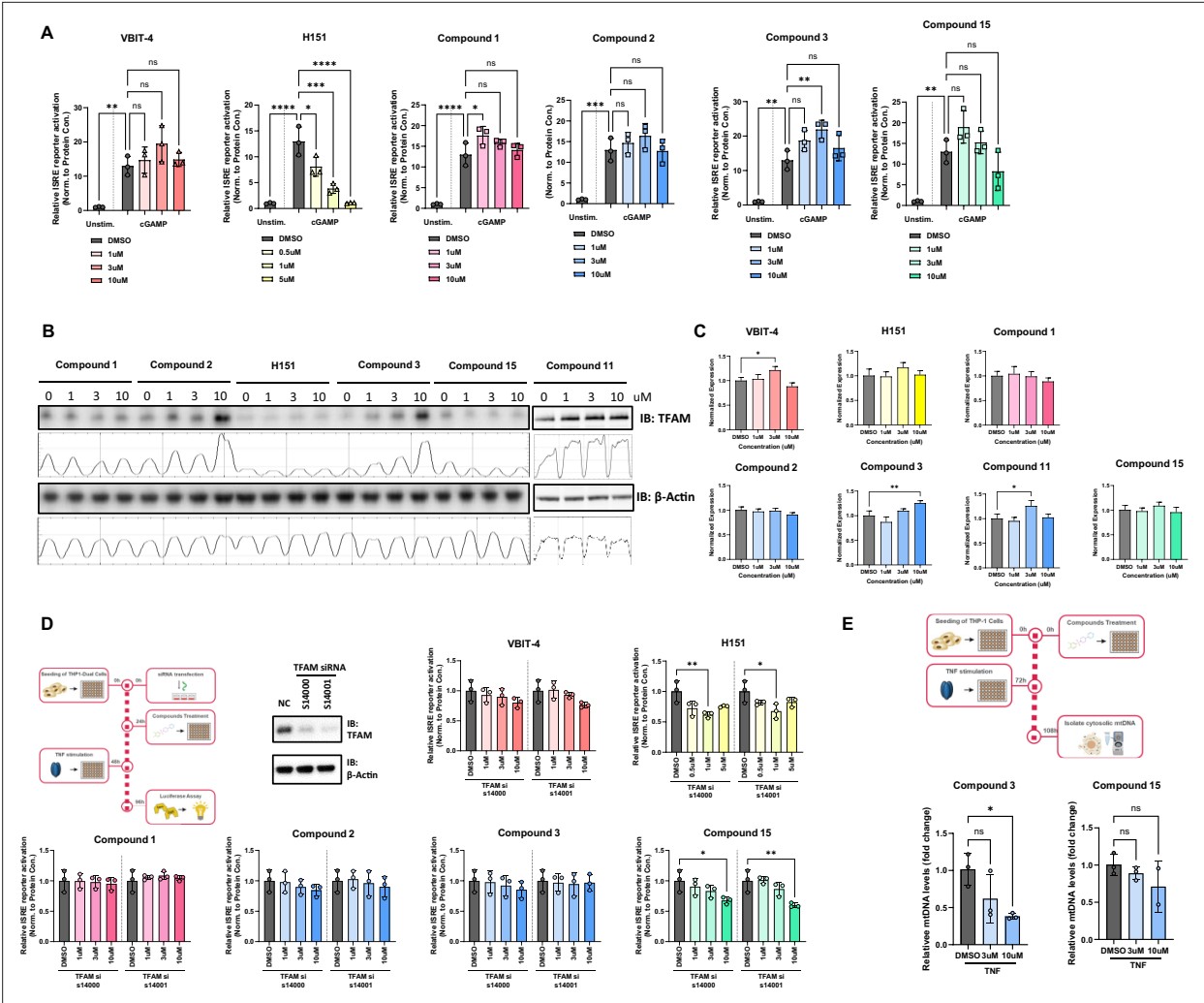

**Figure 4.** Mechanism of action. (**A**) Compounds **2**, **3**, and **15** do not repress cGAMP induced ISG signaling. THP-1-Dual cells were pre-treated with individual compounds 48 h prior to 10 ug/mL cGAMP stimulation for 24 h. ISRE reporter activation was measured. (**B**) Compounds **2**, **3**, and **11** impart a dose-dependent increase in TFAM protein levels. Immunoblot analysis of TFAM from T47D cells treated with indicated compounds for 5 days. (**C**) Compounds exhibit minimal impact on TFAM mRNA levels. (**D**) Downregulation of TFAM attenuates the function of compounds **2** and **3** in repression of ISG signaling. THP-1-Dual cells were treated with individual compounds 24 h after siRNA transfection. After incubation for 24 h, THP-1-Dual cells were stimulated with 100 ng/mL TNFa for another 48 h. ISRE reporter activation was measured and normalized to protein concentration. (**E**) Compound **3** inhibits mtDNA cytosolic leakage. THP-1 cells were pre-treated with individual compounds 72 h prior to 100 ng/mL TNFa stimulation for 48 h. Cytosolic mtDNA was extracted and quantified using a qPCR assay. Figures are representatives of at least two independent experiments. Graph shows one representative experiment of two independent experiments. Error bars represent ± SD from n=3 biological replicates. *p<0.05; **p<0.01; ***p<0.001; ****p<0.0001. Source data for this figure are available in *Figure 4—source data 1* (original uncropped blots) and *Figure 4—source data 2* (annotated uncropped blots), and *Figure 4—source data 3* (raw data and analysis).

The online version of this article includes the following source data for figure 4:

**Source data 1.** Original uncropped western blot images for *Figure 4*.

**Source data 2.** Annotated uncropped western blot images for *Figure 4*, with treatment conditions and protein identities indicated.

**Source data 3.** Raw data and analysis results to generate the graphs shown in *Figure 4*.

abnormality in the respiratory chain production of ATP within the mitochondria (*Fan et al., 2021*). We hypothesize that elevation of TFAM protein levels should increase overall mtDNA levels (mutant and wild-type) and help restore normal mitochondrial function by boosting the absolute levels of wild-type mtDNA. To determine whether our TFAM compounds improved MELAS cellular bioenergetics, we determined their effect on intracellular ATP levels in control and MELAS 80% m.3243G mutant mtDNA cybrid cells (*Bacman et al., 2020*). Treatment with compound **2** caused a significant increase

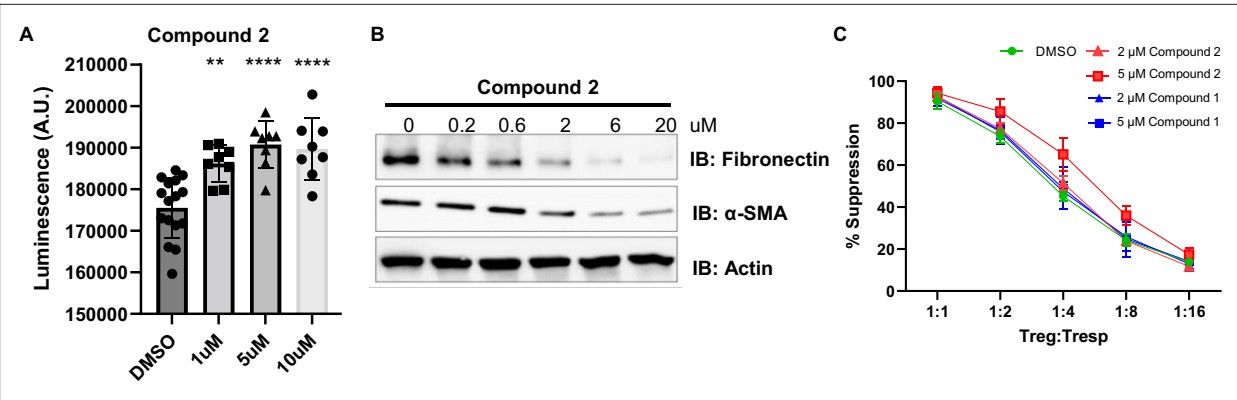

**Figure 5.** Impact of compound **2** in representative cellular disease models. (**A**) Compound **2** dose-dependently increases ATP levels in MELAS 80% cybrid cells after 48 h treatment. Cells were treated with increasing concentrations of Compound **2** or DMSO (vehicle control) for 48 h. ATP levels were measured using the CellTiter-Glo luminescent assay (Promega). Data are presented as mean ± SEM from n=8 biological replicates. Statistical significance was determined using one-way ANOVA followed by Dunnett's multiple comparisons test; *p<0.05; **p<0.01; ***p<0.005; ***p<0.001 compared to DMSO control. (**B**) Western blot analysis Compound **2** dose-dependently decreases fibrotic markers (Fibronectin and α-SMA) of SSc patient fibroblasts after 48 h treatment. Actin is shown as a loading control. (**C**) Compound **2** increases the suppressing capacity of Tregs against effector T cells in a dose-dependent fashion. ITreg cells from four donors stained with Celltrace violet (CTV) are cocultured with anti-CD3/CD28 stimulated autologous CFSE-labeled responder cells (Tresp) for 4 days followed by staining for viability and CD4 and analysis on flow cytometer for dilution of CFSE. The percentage suppression of Tresp cells was calculated for each group with different Treg:Tresp ratio. Source data for Figure 5 are provided in *Figure 5—source data 1* (original uncropped blots) and *Figure 5—source data 2* (annotated uncropped blots).

The online version of this article includes the following source data for figure 5:

**Source data 1.** Original uncropped western blot images for *Figure 5*.

**Source data 2.** Annotated uncropped western blot images for *Figure 5*, with treatment conditions and protein identities indicated.

in ATP levels in MELAS 80% cybrid cells in a dose-dependent fashion (*Figure 5A*), suggesting that TFAM modulation could rescue the energetic defects observed in MELAS.

Additionally, the analysis of kidney fibrosis in both patient and animal models reveals significant mitochondrial defect with a marked downregulation of TFAM in kidney tubule cells. TFAM heterozygous mice show aberrant packaging of mtDNA resulting in escape of mtDNA into the cytosol, activation of the cytosolic cGAS-STING pathway (*Newman and Shadel, 2023*). We investigated the effect of TFAM compounds on the fibrotic pathogenesis in SSc fibroblasts. As shown in *Figure 5B*, fibrotic markers such as fibronectin and α-SMA decrease significantly with the treatment of compound **2**. These preliminary results merit further investigation into TFAM modulation for the treatment of fibrosis.

Additionally, elevation of TFAM protein levels may provide therapeutic benefit for patients suffering from autoimmune diseases. Regulatory T cells (Tregs) possess the ability to potently suppress T cell responses and maintain proper immune homeostasis. Mitochondrial dysfunction resulting from damaged respiratory chain complexes and abnormal mitochondrial morphology can markedly impair the survival, differentiation, and function of Tregs (*Han et al., 2023*). Therefore, increased content of mtDNA in Tregs of patients with autoimmune diseases could be a potential therapeutic strategy. To validate our hypothesis, we treated Tregs with TFAM modulating compounds and measured the suppressing capacity of Tregs against effector T cells. In *Figure 5C*, Teff cells co-cultured with Tregs treated with compound **2** exhibited marked inhibition of proliferation in a dose-dependent manner, which indicates that our TFAM compounds demonstrate an anti-inflammatory response and could suppress the harmful effects of immune responses of autoimmunity.

## Discussion

TFAM is a key regulator of mtDNA maintenance, packaging, and replication (*Kang et al., 2018*). Beyond its essential role in mitochondrial bioenergetics, TFAM has recently been implicated in modulating immune responses through its effects on mtDNA stability (*Li et al., 2024*). When mtDNA is destabilized or released into the cytoplasm due to mitochondrial damage, it can activate the cGAS-STING

pathway, resulting in the induction of ISGs and driving inflammation (*Hu and Shu, 2023*). Previous studies have shown that mitochondrial dysfunction and the subsequent activation of the cGAS-STING pathway contribute to the pathogenesis of various diseases, including autoimmune disorders (*Hu and Shu, 2023*), neurodegenerative diseases (*Huang et al., 2023*), and cancer (*Kwon and Bakhoum, 2020*). Given its central role in maintaining mitochondrial integrity, TFAM represents a promising therapeutic target to mitigate aberrant immune activation and restore cellular homeostasis (*Decout et al., 2021*). However, the development of pharmacological modulators of TFAM has remained largely unexplored until now.

In this study, we successfully identified small-molecule activators of TFAM, demonstrating their ability to enhance TFAM protein levels and modulate mtDNA copy number. From a high-throughput screening approach, we discovered that these compounds also suppress ISG signaling, suggesting their potential to alleviate a cGAS-STING-mediated inflammatory response resulting from mitochondrial damage. Our results demonstrate that molecules such as compound **2** effectively increased TFAM protein levels and activity, thereby promoting mitochondrial integrity and reducing ISG expression. Notably, the observed increase in mtDNA copy number with these compounds remained within a range consistent with safe and beneficial intervention (*Figure 1C and D*), as further supported by improvements seen in cellular disease models (*Figure 5*). Conversely, compound **1** was identified as a TFAM inhibitor, reducing mtDNA copy number. Furthermore, our findings indicate that the suppression of ISG signaling by these activators is likely mediated through the modulation of the cGAS-STING pathway induced by mitochondrial damage. The ability of these compounds to restore mitochondrial function and reduce ISG activation suggests a novel mechanism linking TFAM activation to mitochondrial DNA stabilization and immune signaling regulation.

The suppression of ISG signaling observed in our study aligns with recent findings linking mitochondrial health to immune regulation. The ability of our compounds to enhance mitochondrial function and inhibit an inflammatory response addresses a critical need in treating diseases driven by mitochondrial dysfunction and chronic inflammation. In our study, the identification of TFAM modulators opens new therapeutic opportunities to address diseases characterized by mitochondrial dysfunction, such as MELAS syndrome, fibrosis, and autoimmune disorders (*Figure 5*). Particularly, chemical tools such as compound **2** and related analogs, which enhance both TFAM levels and its activity, represent promising candidates for further scientific exploration.

Enhancing TFAM levels to restore mitochondrial function may also help rescue subtler mitochondrial dysfunction observed in ailments such as Parkinson's and Alzheimer's diseases (*Bustamante-Barrientos et al., 2023*). In addition, dysregulation of TFAM directly leads to altered expression of mtDNA in tumor cells, resulting in cellular metabolic reprogramming and mitochondrial dysfunction. This dysregulation plays a role in modulating tumor progression (*Lei et al., 2024*). Therefore, targeting TFAM may provide potential therapeutic strategies for cancer treatment.

While our study provides strong evidence for the potential of TFAM modulators, further in vivo work is required to validate their efficacy and safety in disease models. Additionally, detailed mechanistic studies are needed to fully elucidate the interplay between TFAM activation and immune signaling pathways. Future work will focus on lead compound optimization and exploring their effects in preclinical models of mitochondrial dysfunction. Additionally, exploring combination therapies with existing mitochondrial-targeting or anti-inflammatory drugs may enhance therapeutic outcomes.

This study highlights the potential of small-molecule TFAM modulators as a therapeutic strategy for diseases driven by mitochondrial dysfunction and aberrant immune activation. By increasing TFAM protein levels, restoring mitochondrial function, and suppressing aberrant immune activation, these compounds offer valuable tools for further exploring the therapeutic potential of targeting TFAM.

# Materials and methods
## Cell lines

THP-1 (TIB-202) and T-47D (HTB-133) cell lines were obtained from ATCC. THP1-Dual (thpd-nfis) cell line was obtained from InvivoGen. THP-1 cells were cultured in RPMI-1640 Medium (Gibco, A10491-01) with 10% FBS (Gibco, 16140-071) with 0.05 mM 2-mercaptoethanol (Sigma, M6250). T-47D cells were cultured in RPMI-1640 Medium (Gibco, A10491-01) with 10% FBS (Gibco, 16140-071) with 0.2 Units/mL insulin (Gibco, 12585014). THP1-Dual cells were cultured in RPMI-1640 Medium (Gibco,

A10491-01) with 10% FBS (Gibco, 16140-071) with 100 µg/mL Normocin (InvivoGen, ant-zn-1), and 100 U/mL Pen-Strep (Gibco, 15-140-122). Primary SSc patient lung fibroblasts were cultured in DMEM (Gibco) according to a previous technique (*Lear et al., 2020*; *Mari and Crestani, 2019*; *Valenzi et al., 2019*). All cell lines were authenticated by STR profiling and routinely tested negative for mycoplasma contamination.

### Regents

VBIT-4 (Selleck Chemicals, s3544); H-151 (InvivoGen, INHH151); TNF-α (Biolegend, 575204); cGAMP (InvivoGen, tlrl-nacga23-02); TFAM siRNAs (Thermo Fisher Scientific, s14000 and s14001).

### mtDNA assay

T-47D cells were treated with test compound at the desired concentration for 5 days, and then washed plates with PBS five times. Isolate DNA following manufacturer instructions using Zymo Quick-DNA 96 Kit. Prepare the PCR reaction using the Bravo Liquid Handler and then place the plate in the PCR system and use the following settings: Experiment properties: 384-well block; Comparative CT ($\Delta\Delta$CT) experiment; SYBR Green Reagents; Fast run (include melting curve). Run method: Step 1 (95°C for 20 s), Step 2 (40 cycles, 95°C for 1 s, 60°C for 20 s), Step 3 (95°C for 15 s, 60°C for 60 s, 95°C for 15 s). Use the average CT values of each duplicate for both genes to calculate $\Delta$CT = CTND1 − CTrRNA. Subtract $\Delta$CT of vehicle sample from test compound sample to calculate $\Delta\Delta$CT = $\Delta$CTcpd − $\Delta$CTveh. Calculate the Relative Quantification RQ = 2-$\Delta\Delta$CT. Plot the RQ values at the different doses of test compound.

### CXCL10 protein detection

THP-1 cells ($1 \times 10^6$/mL) treated with 100 ng/mL TNF-α (Biolegend, 575204) for 48 h, or cGAMP (InvivoGen, tlrl-nacga23-02) for 24 h to induce a cGAS-STING-dependent interferon response. Cells were pretreated with TFAM compounds for 3 days before TNF treatment. 5 uL of the cell-free supernatant was used to measure CXCL10 protein content following the instruction of the CXCL10 AlphaLisa Kit (PerkinElmer AL259C). Signals were measured with an EnVision Multi-mode Plate Reader.

### ISRE luciferase reporter assay

THP-1-Dual cells (thpd-nifs, $1 \times 10^6$/mL) treated with 100 ng/mL TNF (Biolegend, 575204) for 48 h, or cGAMP (InvivoGen, tlrl-nacga23-02) for 24 h to induce a cGAS-STING-dependent interferon response. Cells were pretreated with TFAM compounds for 24 h before TNF treatment. 5 uL cell-free supernatants were mixed with 25 uL QUANTI-Luc substrate (rep-qlc4lg1) in a 384-well opaque plate and luciferase activity was measured for 0.1 s in an EnVision Multimode Plate Reader.

### Compound screen

Compounds were added to 96-well plates after THP1-Dual cells ($2 \times 10^5$ cells/well) were seeded into the wells. 24 h later, cells were stimulated with TNF (100 ng/mL final concentration). 48 h later, ISRE reporter activity was measured as indicated in the section "ISRE luciferase reporter assay'.' Protein concentration was measured using BCA protein assay kit (Pierce, 23225) for normalization. 100% activation was calculated for each plate by dividing the mean of DMSO containing and TNF-treated well with protein concentration. Based on the 100% activation, for each compound, the percent activation was calculated (final concentrations of 40, 20, 10, 5, 2.5, 1.25, 0.625, and 0.3125 uM). For the illustration, a log2 change was calculated.

### Reverse transcription-quantitative PCR analysis (RT-qPCR)

RNA samples prepared with the Maxwell(R) RSC simplyRNA Cells (Promega, AS1390). Reverse transcription was performed with indicated primers using Applied Biosystems High-Capacity cDNA Reverse Transcription Kit (Applied Biosystems, 4374966). Quantitative Real-Time PCR was performed using iQ SYBR Green Supermix (Bio-Rad, 1708880). Relative mRNA expression was calculated using the $\Delta\Delta$CT method.

### Immunoblot

Cells were lysed in RIPA buffer (Abcam, AB15603415ML) plus protease and phosphatase inhibitors (Pierce, PIA32961) for 30 min on ice. Cell lysates were mixed with 4x NuPAGE LDS sample

buffer (InvivoGen, B0007), and boiled for 5 min at 95°C. Protein samples were separated on a 4%–12% NuPAGE Bis-Tris Midi Protein Gels (Invitrogen, WG1403A) and transferred on a PVDF membrane (Bio-Rad, 1704156). Proteins were detected using anti-TFAM polyclonal antibody (Cell Signaling Technology, 7495), and anti-β-actin polyclonal antibody (Invitrogen, PA585271). Primary antibodies were detected using secondary HRP-conjugated goat anti-rabbit antibody (Invitrogen, PI31460). Signals were revealed with clarity ECL substrate (Thermo Scientific, PI34578) in a Bio-Rad ChemiDoc Imaging System.

### siRNA treatment

Lipofectamine RNAiMax (Invitrogen, 13-778-075) was used for transfecting TFAM siRNAs (Thermo Fisher Scientific, s14000 and s14001) as per the manufacturer's instructions. Cells were collected 4 days after transfection.

### Measurement of mitochondrial DNA release

Seed $1 \times 10^6$ THP-1 cells in 5 mL per well into six-well plate in the appropriate medium. At relevant times post-treatment, centrifuge the suspension cells, wash cells with 1× DPBS. Add 100 μL 1% NP-40 (Sigma, IGEPAL CA-630) to each tube. Place lysates into prelabeled microcentrifuge tubes and incubate on ice for 15 min. Spin lysates at 13,000 rpm ($16,000 \times g$) for 15 min at 4°C to pellet the insoluble fraction. Transfer supernatant (the cytosolic fraction) to a new tube and discard the pellet. Use the DNeasy Blood & Tissue Kit (QIAGEN, 69504) to purify mitochondrial DNA from the cytosolic fraction according to the manufacturer's instructions. Add 100 μL ethanol (96–100%) to the cytosolic fraction and continue to step 4 in the DNeasy Blood & Tissue Kit protocol.

Quantitative Real-Time PCR was performed using ABsolute Blue QPCR Mix (Thermo Scientific, AB4322B) for mitochondrial gene Cytochrome c oxidase I (forward: 5′-GCCCCAGATATAGCATTCCC -3′; reverse: 5′-GTTCATCCTGTTCCTGCTCC-3′) and internal control 18S rDNA (forward: 5′-TAGA GGGACAAGTGGCGTTC-3′; reverse: 5′-CGCTGAGCCAGTCAGTGT-3′). Once qPCR is complete, calculate relative fold change in cytochrome c oxidase I from the Ct values.

### Measurement of ATP levels

MELAS 80% m.3243G mutant cybrid cells obtained from Carlos T. Moraes (University of Miami) were seeded at 10k in high glucose media. 24 h post seeding, compound was added in high glucose media via media change. Cells were grown for 48 h prior to running CellTiter-Glo Assay (Promega, G9241).

### Treg suppression assay

Briefly, naive T cells were purified from PBMCs of five healthy donor buffy coats and treated with test substance or DMSO for 6 days in presence of IL-2 (100 U/mL; Peprotech, Ct# 200-02) and TGF-b (5 ng/mL; Biotechne Cat# 240-B-002) in the presence of Dynabeads Human T-activator CD3/28 (Life Technologies, Cat# 11132D) at a ratio of 2:1 (beads:cells). On day 6, induced Treg (iTreg) cells were stained with CellTrace Violet (CTV) (Lifetechnologies Cat# C34557) and were cocultured with anti-CD3/CD28 bead-stimulated autologous CFSE (Life Technologies, Cat# C1157)-labeled responder (Tresp) for an additional 4 days. On day 10, cells were stained for viability and CD4 (Biolegend Cat# 317422) followed by analysis on flow cytometry for dilution of CFSE.

### Quantification and statistical analysis

Except where indicated otherwise, values are reported as the mean ± SD. Statistical significance between groups was calculated using ordinary one-way ANOVA with Sidak multiple comparisons. Further statistical information can be found in the figure legends. In general, $*p<0.05$; $**p<0.01$; $***p<0.001$; and $****p<0.0001$. This is indicated again in the figure legends.

## Acknowledgements

We thank Dr. Carlos T Moraes (The University of Miami Leonard M Miller School of Medicine, Miami, FL, USA) for kindly providing us with the heteroplasmic cell lines. We thank Peter Dwyer for facilitating the transfer of material from the University of Miami. We thank Concept Life Sciences for their support in conducting the Treg study. This research was funded by Astellas.

# Additional information

## Competing interests

Dionisia Sideris: Dionisia Sideris is affiliated with Astellas Pharmaceuticals Inc. The author has no other competing interests to declare. Husan Lee: Hsuan Lee is affiliated with Astellas Pharmaceuticals Inc. The author has no other competing interests to declare. Lyndsay Olson: Lyndsay Olson is affiliated with Astellas Pharmaceuticals Inc. The author has no other competing interests to declare. Kalyan Nallaparaju: Kalyan Nallaparaju is affiliated with Astellas Pharmaceuticals Inc. The author has no other competing interests to declare. Keiichiro Okuyama: Keiichiro Okuyama is affiliated with Astellas Pharmaceuticals Inc. The author has no other competing interests to declare. Jeffrey Ciavarri: Jeff Ciavarri is affiliated with Astellas Pharmaceuticals Inc. The author has no other competing interests to declare. Mads Larsen: Mads Larsen is a founder and stockholder in Coloma Therapeutics. The authors have no other competing interests to declare. Irene Alfaras: Irene Alfaras is a founder and stockholder in Coloma Therapeutics. The authors have no other competing interests to declare. Toren Finkel: Toren Finkel is a co-founder and stockholder in Generian Pharmaceuticals, is a founder and stockholder in Coloma Therapeutics and a scientific advisor for Elangen. The author has no other competing interests to declare. Yuan Liu: Yuan Liu is a co-founder and stockholder in Generian Pharmaceuticals and is a founder and stockholder in Coloma Therapeutics. The author has no other competing interests to declare. Bill Chen: Bill Chen is a co-founder and stockholder in Generian Pharmaceuticals and is a founder and stockholder in Coloma Therapeutics. The author has no other competing interests to declare. Lin Lyu: Lin Lyu is affiliated with Astellas Pharmaceuticals Inc. The author has no other competing interests to declare. The other authors declare that no competing interests exist.

## Funding

| Funder | Grant reference number | Author |
| --- | --- | --- |
| Astellas Pharma (Canada) | Internal funding | Lin Lyu |

The funders had no role in study design, data collection and interpretation, or the decision to submit the work for publication.

## Author contributions

Dionisia Sideris, Conceptualization, Supervision, Writing – original draft, Project administration, Writing – review and editing; Husan Lee, Resources, Data curation, Investigation; Lyndsay Olson, Robert Lafyatis, Mads Larsen, Bo Lin, Irene Alfaras, Jason Kennerdell, Toren Finkel, Yuan Liu, Bill Chen, Data curation, Investigation; Kalyan Nallaparaju, Data curation, Investigation, Methodology, Writing – review and editing; Keiichiro Okuyama, Resources, Investigation, Methodology; Jeffrey Ciavarri, Resources, Methodology, Writing – review and editing; Lin Lyu, Conceptualization, Data curation, Investigation, Methodology, Writing – original draft, Project administration, Writing – review and editing

## Author ORCIDs

Toren Finkel ⓘ https://orcid.org/0000-0002-5265-7982
Bill Chen ⓘ https://orcid.org/0000-0003-2695-5107
Lin Lyu ⓘ https://orcid.org/0000-0001-9771-0110

Reviewer #1 (Public review): https://doi.org/10.7554/eLife.108742.2.sa1
Reviewer #2 (Public review): https://doi.org/10.7554/eLife.108742.2.sa2

# Additional files

## Supplementary files

MDAR checklist

## Data availability

All data supporting the findings of this study are provided in the figures, figure supplements, or source data file. The full original uncropped blot images for all figures are included as source data in this submission. No custom code was used in this study.

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
