## [Editor Report · eLife Assessment]

Using high-throughput small-molecule screening, this study discloses novel modulators of the mitochondrial transcription factor A (TFAM), a key regulator of mitochondrial function. Reviewers viewed the targeting of TFAM as innovative and the study's conclusions as potentially **important** (especially the effects on inflammation). However, the lack of evidence for a direct effect of the compounds on TFAM activity weakens the paper's key conclusion and renders the study **incomplete**.

---

## [Referee Report · Reviewer #1 (Public review)]

Summary:

The authors identify small-molecule compounds modulating the stability of the mitochondrial transcription factor A (TFAM) using a high-throughput CETSA screen and subsequent secondary assays. The identified compounds increased the protein levels of TFAM without affecting its RNA levels and led to an increase in mtDNA levels. As a read-out for dose-dependent action of the identified compounds, the authors investigated cGAS-STING and ISG activation in cellular inflammation models in the presence or absence of their compounds. The addition of TFAM modulators led to a decrease in cGAS-STING/ISG activation and decreased mtDNA release. Furthermore, beneficial effects could be determined in models of mtDNA disease (rescue of ATP rates), sclerotic fibroblasts (decreased fibrosis), and regulatory T cells (decreased activation of effector T cells). The study thus proposes novel first-in-class regulators of TFAM as a therapeutic option in conditions of mitochondrial dysfunction.

Strengths:

The authors identified TFAM as a promising target in conditions of mitochondrial dysfunction, as it is a key regulator of mitochondrial function, serving both as a transcription and packaging factor of mtDNA. Importantly, TFAM is a key regulator of mtDNA copy number, and a moderate increase in TFAM/mtDNA levels has been shown to be beneficial in a number of pathological conditions. Furthermore, mtDNA release leading to activation of inflammatory responses has been linked to a variety of pathological conditions in the last decade. Thus, the identification of small molecule modulators of TFAM that have the potential to increase mtDNA copy number and decrease inflammatory signaling is of great importance. Furthermore, the authors highlight potential applications in the field of mitochondrial disease, fibrosis, and autoimmune disease.

Weaknesses:

The central weakness of the study is the fact that the authors propose compounds as modulators or even activators of TFAM without sufficiently proving a direct effect on TFAM itself. There are no data indicating a direct effect on TFAM activity (e.g., mtDNA transcription, replication, packaging), and it is not sufficiently ruled out that other proteins (e.g., LONP1) mediate the effect. Additionally, important information on the performed screen is not provided. Thus, the data presented is currently incomplete to support the described findings. Furthermore, the introduction and discussion are lacking key references.

---

## [Referee Report · Reviewer #2 (Public review)]

Summary:

The present paper aims to identify small molecules that could possibly affect mitochondrial DNA (mtDNA) stability, limiting cytosolic mtDNA abundance and activation of interferon signaling. The authors developed a high-throughput screen incorporating HiBiT technology to identify possible target compounds affecting mitochondrial transcription factor A (TFAM) content, a compound known to impact mtDNA stability. Cells were subsequently exposed to target compounds to investigate the impact on TNFα-stimulated interferon signaling, a process activated by cytosolic mtDNA abundance. Compound 2, an analog of arylsulfonamide, was highlighted as a possible mitochondrial transcription factor A (TFAM)-activator, and emphasized as a small molecule that could stabilize mtDNA and prevent stress-induced interferon signaling.

Strengths:

Identifying compounds that positively affect mitochondrial biology has diverse implications. The combination of high-throughput screening and assay development to connect identified compounds with cellular interferon signalling events is a strength of the current approach, and the authors should be commended for identifying compounds that broadly impact interferon signalling. The authors have incorporated diverse measurements, including TFAM content, mtDNA content, interferon signaling, and ATP content, as well as verified the necessity of TFAM in mediating the beneficial effects of the emphasized small molecule (Compound 2).

Weaknesses:

(1) While the identified compound clearly works through TFAM, Compound 2 was identified as an arylsulfonamide, which would be expected to affect voltage-gated sodium channels (e.g. PMID: 31316182). Alterations in cellular sodium content and membrane polarization could affect metabolism to indirectly influence mtDNA and TFAM content. It remains unclear if this compound directly or indirectly affects TFAM content, especially as the authors have utilized various cancer cell lines, which could have aberrant sodium channels.

(2) TFAM is nuclear encoded - if this compound directly functions to 'activate TFAM', why/how would TFAM content increase independent of nuclear transcription?

(3) While a listed strength is the incorporation of diverse readouts, this is also a weakness, as there is a lack of consistency between approaches. For instance, data is not provided to show compound 2 increases TFAM or mtDNA content following TNFα stimulation, and extrapolating between cell lines may not be appropriate. The authors are encouraged to directly report TFAM and mtDNA for target compounds 2 and 15 to support their data reported in Figure 2. Ideally, the authors would also report for compound 1 as a control.

(4) While the authors indicate compound 11 displayed the strongest effect on ISRE activity, this appears not to be identified in Figure 1B as a compound affecting TFAM content? Can the authors identify various Compounds in Figure 1B to better highlight the relationship between compounds and TFAM content?

(5) The authors suggest Compound 2 increases cellular ATP - but they are encouraged to normalize luminescence to cellular protein and OXPHOS content to better interpret this data. Additionally, the authors are encouraged to report cellular ATP content following TNFα stimulation/stress (the key emphasis of the present data) and test compound 11, which the authors have implicated as a more sensitive compound.

The discussion is really a perspective, theorizing the diverse implications of small molecule activation of TFAM. The authors are encouraged to provide a balanced discussion, including a critical evaluation of their own work, including an acknowledgement that evidence is not provided that Compound 2 directly activates TFAM or decreases mtDNA cytosolic leakage.